# Present and Future Undergraduate Students' Well-Being: Role of Time Perspective, Self-Efficacy, Self-Regulation and Intention to Drop-Out

Maria Lidia Mascia [1,*], Mirian Agus [1], Cristina Cabras [1], Diego Bellini [2], Roberta Renati [1] and Maria Pietronilla Penna [1]

1 Department of Pedagogy, Psychology, Philosophy, University of Cagliari, 09123 Cagliari, Italy
2 Faculty of Medicine, University of Cagliari, 09042 Cagliari, Italy
* Correspondence: marialidia.mascia@unica.it

**Abstract:** Well-being is a multidimensional construct that affects various areas of a person's life. In the university context, a student's well-being can influence not only their academic and professional success but also the future development of society. This study aimed to evaluate how the interactions of time perspective (assessed by the Stanford Time Perspective Inventory—Short Form), self-efficacy (assessed by the General Self-efficacy Scale), self-regulation (assessed by the Self-regulated Knowledge Scale—University), and drop-out intention (assessed by the Intention to Drop-Out Scale) affect students' perceptions of current and future well-being (assessed by the I COPPE Scale). Using a cross-sectional design, 192 students attending the University of Cagliari (Italy) were evaluated. A partial least squares structural equation modelling (PLS-SEM) analysis was performed to examine the relationships among well-being and all the variables examined herein. The empirical findings highlighted the direct and indirect effects of the studied variables on students' current and future well-being.

**Keywords:** university students; well-being; dropout; time perspective; self-efficacy; self-regulation

## 1. Introduction

The existing literature on well-being defines it as a multidimensional and holistic construct comprised of a set of several intersecting factors and dimensions [1–3]. The multidimensional aspect of well-being is visible in the models presented in the literature. Researchers agree that a combination of internal and external factors and conditions makes an individual feel good [4]. The World Health Organization considers well-being to be the experience of positive emotions, such as happiness and fulfilment, the development of one's potential, control over one's life, a sense of purpose, and the experience of positive relationships [5]. In the last four decades, researchers have focused on specific aspects of well-being, particularly psychological well-being [6] and subjective well-being [7,8]. The entirety of the research in this field presents a conceptualisation of well-being that is focused on the identification of the individual and contextual factors that lead to satisfying lives, favouring the use of models that consider multiple domains. One such multidimensional model of well-being was proposed by Prilleltensky and colleagues [9]. This model, called I COPPE, considers six relevant life domains: Interpersonal, Community, Occupational, Physical, Psychological, and Economic. Interpersonal well-being refers to a person's degree of satisfaction with their intimate relationships with their family, friends, and colleagues. Community well-being refers to a person's satisfaction with the place where they live. Occupational well-being refers to a person's level of satisfaction with their main activity, such as working or caring for their home and family. Physical well-being refers to a person's general state of health. Psychological well-being refers to a person's degree of satisfaction with their emotional life. Economic well-being refers to a person's financial situation [9].

The I COPPE model confirms a systemic vision of well-being that can be achieved by the simultaneous fulfilment of needs at the individual, relational, organisational, and community levels [3,9]. Well-being is a dynamic concept that includes subjective, social, and psychological dimensions as well as health-related behaviours [10]. The theme of well-being is investigated across all age groups; however, the present study focused on university students. The well-being of young people, such as students, is a concern at both the national and the international level; furthermore, it is increasingly at the centre of policies, programmes, and the professional development of teachers in schools. Support for well-being is a growing topic, as evidenced by the increasing amount of literature highlighting the expansion of participation in specific programmes that promote well-being [11]. At university, students encounter a new context that is markedly different from that of high school; thus, higher education often focuses on promoting the positive and holistic development of university students [12]. Moreover, concerns about student well-being became prominent during the COVID-19 pandemic, which highlighted various issues faced by students [13–17].

## 2. The Present Study

The context of education is an important environment that offers opportunities for personal growth [17] and professional success. Therefore, the present study aimed to develop an explanatory model to determine the factors that may contribute to present and future well-being among university students, with the objective of improving university conditions through the identification of effective interventions or changes, primarily in the Italian context. The need to find reference models has also been emphasised in recent reviews and meta-analyses, questioning what can be considered the antecedents of well-being among university students [18,19]. The factors involved in university well-being are numerous; the problem is highly complex as there are several inter-related factors that may be both positive and negative antecedents of well-being. The present study was intended to investigate the empirical links among the variables identified in the literature as the antecedents of academic success as well as academic and professional well-being.

### 2.1. Future Time Perspective Related to Self-Regulation, Self-Efficacy, Intention to Dropout, and Well-Being

The theory of Time Perspective (TP) conceptualised by Zimbardo [20], assumes that TP is situationally determined and differs among individuals. TP can be considered an often-unconscious process whereby the continual flows of personal and social experiences are assigned to temporal categories or time frames that help give order, coherence, and meaning to those events. This theory proposes that a person's motivational, emotional, behavioural, cognitive, and social processes are influenced by the way he/she sees the past, present, and future [20]. Time perspective is learned and determined by multiple factors, such as culture, education, social class, age, and more. According to Zimbardo [20], individuals develop an overreliance on a particular time frame that operates, most of the time, on an unconscious level and influences much of their judgments, decisions, and actions. For example, people who over-rely on the future temporal frame are more likely to be risk-aversive, have higher grades and make healthier choices than those who are more present-oriented because they are thinking about the future consequences of their decisions [20]. Thus, the future time perspective (FTP), the individual's perception of his/her remaining time to live, has been a focus of growing interest in psychology over the past decade, especially in the fields of aging and health [21]. Time is not just a physical phenomenon; it is open to psychological interpretation, according to James [22]. FTP represents an important psychological variable that can be traced back to Lewin [23], who claimed that a person's life-space includes not only a geographical and a social environment, but also a temporal dimension. Lewin [23] stated that FTP influences a person's behaviour and asserted that change within FTP is one of the most fundamental facts of development. Denovan and colleagues [24] emphasised the correlation among FTP, positive emotion and student engagement.

An individual's TP may impact other constructs, such as achievement, goal setting, addiction, rumination, and well-being [25]. Avci [26] found that FTP and self-regulation strategies have a positive effect on first-grade university students; they could set and attain goals concerning their academic activity more easily. This relationship has been confirmed by other studies and meta-analyses [27–29].

Gutiérrez-Braojos [27] presented a notable and complete analysis of the direct and indirect relationship between FTP and self-efficacy in academic achievement. In particular, he reported on the research by Chung et al. [30], which showed that students who linked their future professions to their academic studies exhibited higher self-efficacy beliefs in learning course content and better academic success compared to students without FTP. The relationship between FTP and self-efficacy has been confirmed not only in the university context but also regarding the motivational role of FTP as adaptivity in career construction [31].

Furthermore, FTP appears to be related to motivation and educational drop-out; the latter aspect is associated with people with a short FPT [32].

FTP is related to positive goals and the development of well-being [33]. The relation between FTP and well-being is confirmed in a recent meta-analysis by Kooij and colleagues [34]; however, the ability to foresee, anticipate and plan for future desired outcomes is crucial for present and future well-being. Li Wen Chua and colleagues [35] showed that FTP has important implications for the health and well-being of adolescents. Mascia and colleagues [36] confirmed the positive implications of this related to academic achievement among university students.

### 2.2. Self-Efficacy Related to Self-Regulation, Intention to Drop-Out, and Well-Being

Self-efficacy is a universal psychological need that controls an individual's cognition, emotions, and decisions related to psychological well-being [37]. Compared to people with poor self-efficacy, those with high self-efficacy are more likely to use highly adaptive coping mechanisms [38,39]. The relationship between self-efficacy and self-regulation is strongly supported by literature; low self-efficacy and self-regulation can be predictors of drop-out intention [40,41]. Sabouripour and colleagues [42] emphasise a mediation role between optimism, dimensions of psychological well-being, and resilience among Iranian students. Self-efficacy may play a role in how students feel about themselves and whether they accomplish their goals effectively in life. The association between optimism, psychological well-being, and resilience has been confirmed in several studies. Hence, self-efficacy is an essential personal resource for university students to prevent stressors and promote adaptive adjustment [43]. Self-efficacy has been shown to predict students' scholastic performance and success. As reported by some research studies, students who express high levels of self-efficacy and well-being are motivated to participate in relevant academic activities and to develop positive attitudes that lead to success at school [44,45]. They are also likely to perform well in achieving their academic goals [46]. He and colleagues [47] showed that self-efficacy had a positive effect on psychological well-being among nursing students.

### 2.3. Self-Regulation Related to Intention to Drop-Out and Well-Being

Research on university norms suggests that personal and environmental characteristics might elicit such fit effects in undergraduate students. Compared to other educational institutions, universities demand increased initiative and self-regulation from students for them to succeed [48]. Many studies confirmed the positive influence of a good level of self-regulation on well-being and academic retention and success [49,50]. The literature shows that highly self-regulated students know what they want to learn; they plan and control their own learning process using the strategies most suitable for this purpose, they monitor their results and, if necessary, they redefine or modify their goals based on what they have experienced [50–52]. University students are expected to develop critical thinking skills and strengthen their cognitive skills through behaviours, such as perseverance, self-discipline,

self-regulation, and motivation; in order to face the difficulties, they encounter in their academic achievement process [16,53]. High self-regulation levels are positively correlated with lower stress levels among university students [54]; high self-regulation levels are correlated with high well-being levels among the general population [55]. However, self-regulation is considered one of the most important qualities of human beings, as it has allowed us to survive and progress considerably over the ages [56]. Self-regulation skills are significantly related to positive aspects (e.g., lower depression, anxiety, and stress) among the general population and among students [57,58]. In general, self-regulation and self-control influence the well-being of undergraduates [59,60].

*2.4. Academic Achievement, Drop-Out, and Well-Being*

In the scenario of the university system, students' well-being can be undermined by specific elements that, if not controlled or managed, can lead to forms of malaise that can affect their current/present and future well-being. It is also important to understand how a student fits into a university context, particularly in terms of his/her goals and academic retention as well as access to individual and contextual resources. A signal of the student's malaise may be his/her intention to drop-out. In the literature, the term drop-out refers to a student that leaves his/her university studies before having completed the study programme and obtaining a degree [61]. A student drop-out is generated by a long decision-making process and the complex interaction between several factors [62]. The drop-out intention is the result of a complex, dynamic, cumulative, and multifactorial process. This is also emphasised in a review of the literature [63], which highlights the complex interaction among individual, organisational, and social factors that come into play in this process [64,65]. This can be distinguished by the students' motives to voluntarily abandon their studies [66]. Many studies [36,67] confirmed that the intention to drop-out and dropping out are present in the period of transition from high school to university, particularly in the Italian school system, where the two systems are organised very differently; if not well managed, this transition leads to destabilisation in the student's life. In empirical research, many of these determinants indicating dropping out voluntarily or involuntarily are analysed. A recent review of 44 empirical studies on student drop-outs from universities in Europe [68] identified nine main factors that influence the decision to drop-out or transfer to a different study programme, subject, or university: study conditions at university; academic integration at university; social integration at university; personal effort and motivation for studying; information and admissions requirements; prior academic achievement in school; personal characteristics of the student; sociodemographic background of the student; external conditions. In this scenario, it is essential to understand the risk and protective factors, the antecedents, and the consequences that these factors have on student well-being. Although some theories support approaching education disruptions as having potentially diverse meanings and effects for some well-being dimensions and facets [69]. In this work, among our hypothesises we look for verifying if the intention to drop-out can have a negative correlation on present and future well-being, starting from the idea that belonging to the university and academic achievement are key variables in promoting students' well-being [48].

*2.5. Aim and Hypotheses*

On the basis of the literature taken into account, the following hypotheses have guided our study, in order to identify predictors of students' psychological present and future well-being (see Figure 1).

The first hypothesis is:

**H1.** *Future perspective might affect (a) self-regulation, (b) self-efficacy, (c) intention to drop-out, (d) present well-being, and (e) future well-being.*

Thus, the second hypothesis of this study is:

**H2.** *Self-efficacy might affect the (a) intention to drop-out, (b) self-regulation, (c) present well-being, and (d) future well-being.*

Then, the third hypothesis of this work is:

**H3.** *Self-regulation might affect the (a) intention to drop-out, (b) present, and (c) future well-being.*

Formerly, the fourth hypothesis of this research is:

**H4.** *Intention to drop-out might affect (a) present well-being, and (b) future well-being.*

Before, the fifth hypothesis of this study is:

**H5.** *The present well-being might affect the (a) future well-being.*

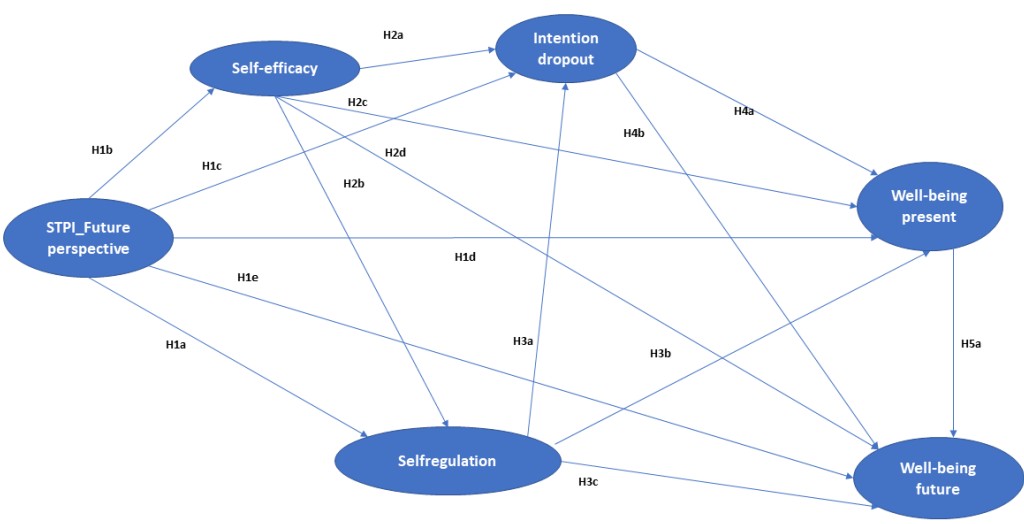

**Figure 1.** Conceptual framework and hypothesis.

### 3. Materials and Methods

*3.1. Participants Recruitment*

The participants were recruited by a non-probabilistic sampling, based on voluntary participation, and they did not receive any incentive for their involvement; the enrollment in the research was publicised and disseminated during the academic activities at the University of Cagliari (Italy). The protocol of research was approved by the local Ethical Committee.

*3.2. Procedures and Measures*

This study applied a cross-sectional design; quantitative data were collected from an online survey (administered by the LimeSurvey platform) between October 2021 and November 2021. Participants signed an informed consent to participate, in accordance with requirements of the Italian and European norms regarding survey administration and management. To be included in the research, participants must be enrolled at Cagliari University.

The measures administered were articulated in different sections.

The first one included a social, demographic, and academic section (inquiring about age, gender, course, and type of degree attended).

The subsequent sections included different assessment-validated instruments.

3.2.1. Stanford Time Perspective Inventory (STPI) Short Form

The Zimbardo's Stanford Time Perspective Inventory (STPI) Short Form [70] was also administered in their Italian version [71]. This short version, composed of 22 items, considers the present dimensions and the future dimension. We choose to use in our work

only the future dimensions; the items were evaluated by a Likert scale (from 1 to 5). This instrument in the future dimension has good Cronbach's Alpha reliability ($\alpha = 0.67$).

### 3.2.2. General Self-Efficacy Scale

The Italian version of the General Self Efficacy Scale [72] assesses the perceived self-efficacy to predict coping as adaptation after stressful experiences (e.g., "If someone opposes me, I can find the means and ways to get what I want"; "When I am confronted with a problem, I can usually find several solutions"). The questions were evaluated on a 5-point Likert scale extending from 1 (not at all true) to 5 (exactly true); the Cronbach's Alpha reliability was $\alpha = 0.86$.

### 3.2.3. Self-Regulation

The Self-regulated Knowledge Scale University [73] was characterised by 15 items organised into five subscales that evaluate: knowledge networking, knowledge extraction, knowledge practice, knowledge critique, and knowledge monitoring. Each of the subscales consists of three items that define the potential answers to the question "When you study, how often do you do the following things?" The response was rated on a 5-point Likert-type scale, ranging from 1 (never) to 5 (very often).

This instrument has shown good reliability (Cronbach's alphas of the subscales extended from 0.73 to 0.80).

### 3.2.4. Intention to Drop-Out

This scale measures the intentions to persist in, versus drop-out of school. It is composed of three items. Two items are the same used by Vallerand et al. [74], which were "I sometimes consider dropping out of school" and "I intend to drop-out of school". The third item, added by Hardre and Reeve [75] asks about intentions to continue one's schooling: "I sometimes feel unsure about continuing my studies year after year". Authors measure correlation among the three items, so they used a three-item scale, because it allowed us to increase both the scope and reliability of our outcome measure (0.79). The questions were evaluated on a 7-point Likert scale, from 1 (not at all) to 7 (very much so).

### 3.2.5. Coppe—I

Prilleltensky and colleagues [9] integrate different models and aspects of subjective well-being into the I COPPE Scale, devised to evaluate individual perceptions of multidimensional well-being. This instrument was validated in this short form in Italy by Esposito et al. [3]. This scale consisted of 14 items and showed good psychometric features and values of reliability (ranging from a minimum of 0.723, to a maximum of 0.935). The items were related to specific dimensions, referring to two different time periods: present (pr) and future (fu). The dimensions considered were overall well-being, interpersonal well-being, community well-being, occupational well-being, physical well-being, psychological well-being, and economic well-being. The responses to each question were assessed by a scale extending from 0 (worst your life can be) to 10 (best your life can be). In our research, we considered the relationships between present and future dimensions.

### 3.3. Statistics

The descriptive data analyses were carried out to evaluate the distribution of all assessed variables.

In order to explore the relationships among the latent constructs, the Partial Least Squares–Structural Equation Modeling (PLS-SEM) was applied. The aim of the Structural Equation Modeling technique is to define the estimations of relationships between unobserved (latent) variables, measured by specific indicators–items. The PLS-SEM aims to maximise the variance explicated between the latent constructs and it is chosen in a special way, contrasting with variance-based SEM, because might deal efficiently with non-normal data, such as the present research [76]. Specifically in this work, the application of PLS-SEM

was useful regarding the application of explorative research and targeting test complex models [76–78]. Furthermore, the PLS-SEM is advantageous because it might be applied to samples with limited sizes [79–81].

In our work, as defined in the literature, a two-stage analysis approach in the PLS-SEM was applied [79].

Indeed, in the first stage of PLS-SEM we evaluated the measurement model by inspecting the reliability and validity of constructs. This step is decisive for the establishment of the measurement integrity of the latent constructs and must be conducted former the assessment of the inner-structural model [76]. To deeply assess the quality of the measurement model, Dijkstra-Henseler's Rho A, Cronbach's alpha, average variance extracted (AVE), and Adjusted $R^2$ were considered (the thresholds of the values are reported in the "Results" section) [76]. Moreover, the discriminant validity was evaluated by the computation of the Heterotrait–monotrait (HTMT) ratio of correlation. To assess for multicollinearity questions, the variance inflation factors (VIF) for all variables were inspected (the thresholds of the values are described in the "Results" paragraph) [76].

It is useful to consider that in our measurement model, to assess the latent variable of Zimbardo 's STPI Time future perspective, the items belonging to this scale were considered as observed variables [70]; to assess the latent variable of self-efficacy the items of this scale were used as observed variables [72]; for the dimension related to the intention to drop-out as observed variables, we used the items belonging to this dimension [75]. The subscales related to the dimensions of self-regulation [73] were used as observed variables. Finally, to assess the latent variable of the present well-being and future well-being, the subscales of the questionnaire regarding each time perspective were considered [9].

Then, the second step of PLS-SEM application was characterised by the evaluation of the structural model, aiming to assess the relationship between the latent constructs. This is defined as inner model, which was studied by the assessment of all identified pathways among the latent constructs. Specifically, we measured the strength of direct and indirect effects, observing the significance of the path coefficients (settling alpha <0.05); in this step a resampling bootstrapping technique (5000 resamples) was applied.

The structural model is presented in Figure 1.

Data were analysed by the open-source software Jasp [82] and the SmartPLS software, version 3.3.9 [78].

## 4. Results

### 4.1. Participants Characteristics

Out of 192 participants, n = 33 (17.2%) of the students are male; overall, the participants were aged from 18 to 62 years (age *m* = 25.70; *sd* = 9.09).

Of the undergraduates, in the past, 140 (73%) attended high schools having a humanistic curriculum; 38 (20%) attended high schools with scientific curricula; and the remaining 14 (7%) attended technical institutes. The participants at the time of the survey were joining university courses leading to Bachelor's (79.2%) and Master's degrees (20.8%) in the Faculty of Humanistic Studies at the University of Cagliari (Italy).

The descriptive statistics regarding all variables were evaluated (see Table 1).

### 4.2. PLS-SEM Measurement Model

Referring to the measurement model of our variables, the reliability (by Dijkstra-Henseler's rho A and Cronbach's Alpha, which should be higher than 0.70) [83], and validity were evaluated. The observed variables loadings and the Average Variance Extracted (AVE) were considered to evaluate the discriminant and convergent validity (a good convergent validity was observed when indicators' loadings and variables' AVEs are respectively higher than 0.70 and 0.50) [84] (Table 2).

**Table 1.** Descriptive statistics of evaluated variables (n = 192).

| Variables | Mean (SD) |
|---|---|
| How true do you consider the following statements to be from 1 (not at all true) to 7 (extremely true)? | |
| I sometimes think about dropping out of university | 2.67 (1.93) |
| I intend to drop-out of university | 1.40 (0.98) |
| Every year I consider dropping out of university | 1.77 (1.54) |
| STPI Future perspective | 30.67 (6.32) |
| Self-efficacy Scale | 36.13 (7.62) |
| Self-regulation—knowledge extraction | 3.87 (0.96) |
| Self-regulation—knowledge networking | 3.53 (1.06) |
| Self-regulation—knowledge practice | 4.10 (0.90) |
| Self-regulation—knowledge critique | 3.29 (1.00) |
| Self-regulation—knowledge monitoring | 4.30 (0.71) |
| Coppe_Present well-being | 40.36 (12.80) |
| Coppe_Future well-being | 46.23 (12.79) |
| Coppe_Overall well-being | 13.03 (4.08) |
| Coppe_Interpersonal well-being | 14.43 (4.44) |
| Coppe_Community well-being | 11.04 (4.39) |
| Coppe_Occupation well-being | 12.79 (4.67) |
| Coppe_Phisical well-being | 13.02 (4.67) |
| Coppe_Psychological well-being | 11.56 (4.46) |
| Coppe_Economical well-being | 12.23 (4.29) |

**Table 2.** PLS-SEM: Measurement—Outer model.

| Construct | Latent Variable Loadings | Dijkstra-Henseler's Rho A | Cronbach's Alpha | Average Variance Extracted (AVE) | Adjusted $R^2$ |
|---|---|---|---|---|---|
| Self regulation | From 0.549 to 0.708 | 0.630 | 0.604 | 0.389 | 0.278 |
| Intention to drop-out | From 0.746 to 0.926 | 0.903 | 0.822 | 0.733 | 0.120 |
| Self-efficacy | From 0.756 to 0.838 | 0.933 | 0.925 | 0.602 | 0.235 |
| STPI Future Temporal perspective | From 0.408 to 0.744 | 0.844 | 0.817 | 0.419 | |
| Well-being present | From 0.698 to 0.865 | 0.882 | 0.877 | 0.582 | 0.243 |
| Well-being future | From 0.696 to 0.869 | 0.893 | 0.887 | 0.604 | 0.754 |

To evaluate the discriminant validity, we considered also the Heterotrait-monotrait (HTMT) ratio of correlation. HTMT values close to 1 represent a deficiency of discriminant validity. Specifically, we assume that if the value of the HTMT is higher than 0.90 [85,86], there is a lack of discriminant validity. Regarding our data, only the index regarding the measurement of well-being in present and in future perspectives showed values of HTMT over the threshold, but this might be specifically related to the features of our assessment (Table 3).

**Table 3.** Heterotrait-monotrait (HTMT) ratio of correlation.

| | Self_Efficacy | Well Being Future | Well Being Present | STPI | Intention to Drop-Out | Self Regulation |
|---|---|---|---|---|---|---|
| Well being future | 0.405 | | | | | |
| Well being present | 0.378 | 0.973 | | | | |
| STPI | 0.506 | 0.416 | 0.430 | | | |
| Intention to dropout | 0.227 | 0.335 | 0.356 | 0.331 | | |
| Self regulation | 0.321 | 0.203 | 0.200 | 0.713 | 0.139 | |

Furthermore, to check for multicollinearity issues, the variance inflation factors (VIF) for all variables were examined (their values should not be overhead the 5.0 threshold) (Table 4).

**Table 4.** Variance inflation factors (VIF)—Internal values.

| | Well Being Future | Well Being Present | Intention to Drop-Out | Self Regulation |
|---|---|---|---|---|
| Self_efficacy | 1.349 | 1.301 | 1.289 | 1.288 |
| Well being present | 1.282 | | | |
| STPI | 1.826 | 1.744 | 1.648 | 1.288 |
| Intention to drop-out | 1.167 | 1.112 | | |
| Self regulation | 1.367 | 1.362 | 1.349 | |

All results related to the evaluation of the outer model (presented in Tables 2–4) suggest that the measurement quality meets the criteria required in the literature.

### 4.3. PLS-SEM Structural Model

The relationships between latent constructs were evaluated in the structural model by the resampling bootstrapping technique (5000 resamples) of 192 participants, which permitted the evaluation of the significance of path coefficients [77].

The relevant and significant findings of the inner model are shown in Figure 2 and in Table 5 (only the significant results are shown in Table 5; the complete table with all the outcomes obtained in the evaluation of the model can be found in Appendix A).

Specifically, in Figure 2, the paths and the in-line values prove the effects between the variables and their beta coefficients (including their *p*-values).

The predictive power of the model was evaluated with R Squared ($R^2$) values; these values demonstrate the explained variance of endogenous latent variables for the structural model (these values are displayed regarding the endogenous variables).

The $R^2$ of our variables ranged from a minimum of 0.120 (for the Intention to drop-out) to 0.754 (for the well-being future) (see Table 2) [83].

The effect size $F^2$ was appraised to inspect the impact of an independent latent variable on a dependent latent variable [87,88]. The findings highlighted the significant $F^2$ of: Zimbardo Future perspective on self-efficacy (t = 2.707; *p* = 0.007); Zimbardo Future perspective on self-regulation (t= 2.916; *p* = 0.004); well-being present on well-being future (t = 3.922; *p* < 0.001).

The remaining $F^2$ estimated were not significant.

Concerning the structural model (see Table 5), we evaluate the effects referring to each hypothesis.

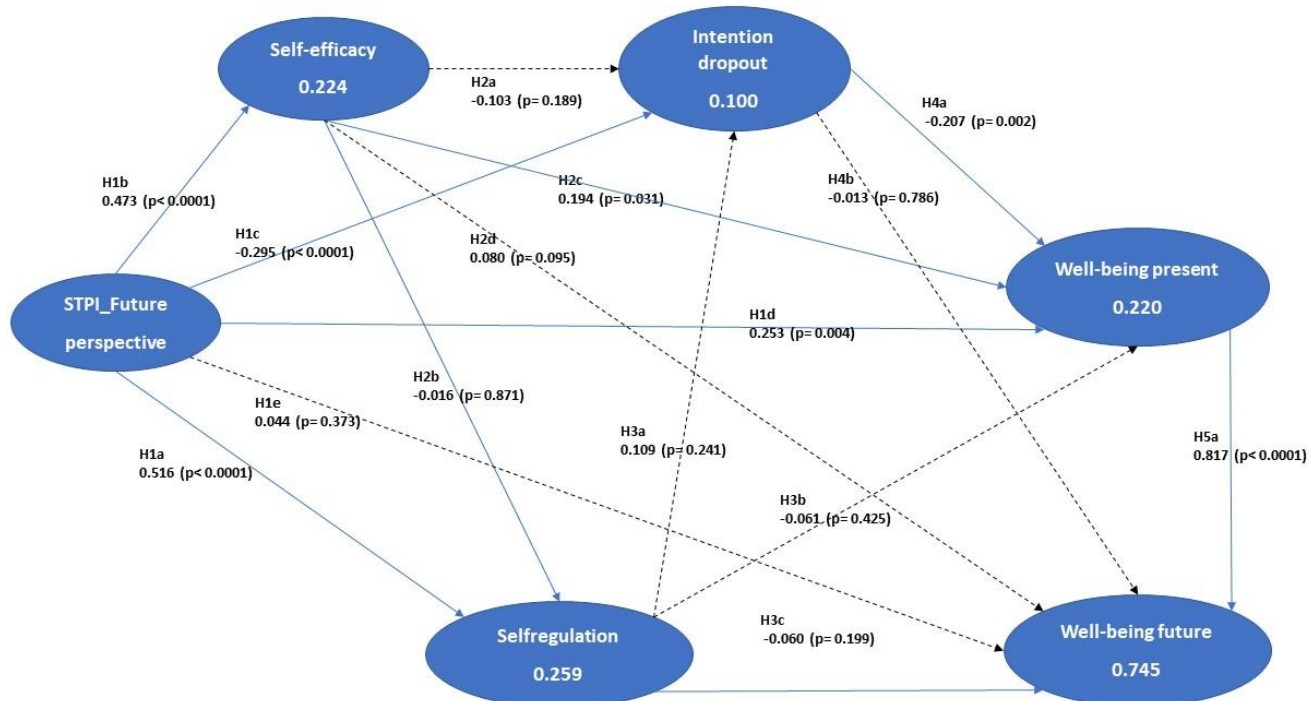

**Figure 2.** SEM-PLS application (Bootstrap method). Note: Non-significant paths ($p > 0.05$) are represented with dotted lines.

**Table 5.** PLS-SEM: Significant paths in the Inner model.

| Hypothesis | Relationship | Standardised Beta | Standard Deviation | T-Value | $p$ | Decision |
|---|---|---|---|---|---|---|
| | DIRECT PATH COEFFICIENT | | | | | |
| H1A | STPI → Self regulation | 0.524 | 0.065 | 7.980 | <0.001 | Supported |
| H1B | STPI → Self efficacy | 0.481 | 0.063 | 7.461 | <0.001 | Supported |
| H1C | STPI → Intention drop-out | −0.302 | 0.082 | 3.592 | <0.001 | Supported |
| H1D | STPI → Wellb_pres | 0.252 | 0.089 | 2.842 | 0.004 | Supported |
| H2C | Self_efficacy → Wellb_pres | 0.193 | 0.090 | 2.163 | 0.031 | Supported |
| H4A | Intention drop-out → Wellb_pres | −0.211 | 0.068 | 3.061 | 0.002 | Supported |
| H5A | Wellb_pres → Wellb_fut | 0.817 | 0.040 | 2.475 | <0.001 | Supported |
| | TOTAL INDIRECT EFFECT | | | | | |
| H1 | STPI → Wellb_fut | 0.323 | 0.065 | 4.895 | <0.001 | Supported |
| H2 | Self_efficacy → Wellb_fut | 0.179 | 0.075 | 2.397 | 0.017 | Supported |
| H4 | Intention drop-out → Wellb_fut | −0.172 | 0.055 | 3.098 | 0.002 | Supported |
| | SPECIFIC INDIRECT EFFECT | | | | | |
| H1 | STPI → Self_efficacy → Wellb_pres | 0.093 | 0.046 | 1.994 | 0.046 | Supported |
| H1 | STPI → Intention drop-out → Wellb_pres → Wellb_fut | 0.052 | 0.023 | 2.192 | 0.028 | Supported |
| H1 | STPI → Wellb_pres → Wellb_fut | 0.206 | 0.074 | 2.810 | 0.005 | Supported |
| H1 | STPI → Intention drop-out → Wellb_pres | 0.064 | 0.028 | 2.164 | 0.031 | Supported |

**Table 5.** *Cont.*

| Hypothesis | Relationship | Standardised Beta | Standard Deviation | T-Value | *p* | Decision |
|---|---|---|---|---|---|---|
| H1 | STPI → Self_efficacy → Wellb_pres → Wellb_fut | 0.076 | 0.038 | 1.999 | 0.046 | Supported |
| H2 | Self_efficacy → Wellb_pres → Wellb_fut | 0.157 | 0.073 | 2.160 | 0.031 | Supported |
| H4 | Intention drop out → Wellb_pres → Wellb_fut | −0.172 | 0.055 | 3.098 | 0.002 | Supported |
| | TOTAL EFFECT | | | | | |
| H1 | STPI → Wellb_fut | 0.366 | 0.072 | 5.036 | <0.001 | Supported |
| H1 | STPI → Wellb_pres | 0.380 | 0.066 | 5.700 | <0.001 | Supported |
| H1 | STPI → Intention drop-out | −0.298 | 0.064 | 4.487 | <0.001 | Supported |
| H1 | STPI → Self regulation | 0.517 | 0.054 | 9.358 | <0.001 | Supported |
| H2 | Self_efficacy → Wellb_fut | 0.258 | 0.095 | 2.735 | 0.006 | Supported |
| H2 | Self_efficacy → Wellb_pres | 0.217 | 0.090 | 2.397 | 0.017 | Supported |
| H4 | Intention drop-out → Wellb_fut | −0.186 | 0.068 | 2.668 | 0.008 | Supported |

Note: H1a: hypothesis 1a; H1b: hypothesis 1b; H2a: hypothesis 2a; H2b: hypothesis 2b; H3a: hypothesis 3a, H3b: hypothesis 3b; p: probability; STPI: Zimbardo's Stanford Time Perspective Inventory; Wellb_pres: Well being present; Wellb_fut: Well being future.

Regarding the H1, the findings identify that Zimbardo's future perspective has a positive significant effect on self-regulation (H1a; β = 0.524, $p$ < 0.001), on self-efficacy (H1b; β = 0.481, $p$ < 0.001), a negative significant effect on the intention to drop-out (H1c; β = −0.302, $p$ < 0.001), a positive significant effect on well-being present (H1d; β = 0.251, $p$ < 0.001); the Zimbardo future perspective did not have a significant effect on well-being future (H1e; β = 0.043, $p$ = 0.373).

Referring to the H1, examination of the indirect effects, highlights the effect of Zimbardo's future perspective on well-being future (β = 0.323, $p \leq$ 0.001). We observed significant specific indirect effects from Zimbardo's future perspective:

- To self-efficacy, to well-being present (β = 0.093, $p$ = 0.046);
- To intention to drop-out, to well-being present, to well-being future (β = 0.052, $p$ = 0.028);
- To well-being present, to well-being future (β = 0.206, $p$ = 0.005);
- To intention to drop-out, to well-being present (β = 0.064, $p$ = 0.031);
- To self-efficacy, to well-being present, to well-being future (β = 0.076, $p$ = 0.046).

The total effects from Zimbardo's future perspective on all latent variables in the model are statistically significant (to Self-regulation β = 0.517, $p$ < 0.001; to Intention to drop-out β = −0.298, $p$ < 0.001; to well-being present β = 0.380, $p$ < 0.001; to well-being future β=0.366, $p$ < 0.001).

Regarding the H2, the findings identify that Self-efficacy did not have a significant effect on the intention to drop-out (H2a; β = −0.107, $p$ = 0.189), on self-regulation (H2b; β = −0.018, $p$ = 0.871), have a positive significant effect on well-being present (H2c; β = 0.193, $p$ = 0.031), did not have a significant effect on well-being future (H2d; β = 0.079, $p$ = 0.095).

Referring to the H2, examination of the indirect effects, highlight the effect of Self-efficacy on well-being future (β = 0.179, $p$ = 0.017).

We observed significant specific indirect effects from self-efficacy to well-being present, to well-being future (β = 0.157, $p$ = 0.031).

The total effects from self-efficacy to Self-regulation (β = -0.018, *p* = 0.871), to Intention to drop-out (β = −0.107, *p* = 0.179) are not significant; to well-being present (β = 0.217, *p* = 0.017); to well-being future (β = 0.258, *p* = 0.006) are significant.

About the H3, the results identify that Self-regulation did not have a significant effect on the intention to drop-out (H3a; β = 0.103, *p* = 0.241), on well-being present (H3b; β=−0.056, *p* = 0.425), on well-being future (H3c; β = −0.057, *p* = 0.199).

Referring to the H3, also the examination of the indirect effects did not highlight the significant effect of Self-regulation on well-being present (β = −0.021, *p* = 0.282) and well-being future (β = −0.065, *p* = 0.285).

Furthermore, we did not observe any significant specific indirect effects or total effects.

Concerning the H4 there is a significant path from intention to drop-out to well-being present (H4a; β = −0.211, *p* = 0.002); furthermore, there is a non-significant path from intention to drop-out to well-being future (H4b; β = −0.014, *p* = 0.786).

Similarly, the examination of the indirect effects highlights the significant effect of the intention to drop-out on a well-being future (β = −0.172, *p* = 0.002). The total effects of the intention to drop-out on well-being present (β = −0.211, *p* = 0.002) and well-being future are significant (β = −0.186, *p* = 0.008).

Finally, regarding the H5, we observe a significant path from well-being present to well-being future (H5a; β = 0.817, *p* < 0.001).

## 5. Discussion and Conclusions

In the present study, we aimed to identify well-being perceptions among university students to develop a model that could show the relationship among specific variables. We focused on individual factors, but with the awareness that environmental characteristics are fundamental to enhancing academic life and professional success [88,89]. Specifically, we wanted to find negative or positive antecedents of well-being, as defined in Prilleltensky and colleagues [9] model. Particularly in this historical period of great changes and uncertainties related also to the COVID-19 pandemic, it is crucial to identify the factors that can be useful for supporting future perspectives in young people and strengthening their will to persist in achieving their goals [14,90,91].

In the model developed in this study, the interactions among the identified variables could provide teachers, educators, and practitioners with interesting empirical perspectives that can be used to support university students in the pursuit of their goals. Our results are in line with the literature in identifying the correlations and causal relationships between the variables.

The data show that the dimension related to the future perspective has a statistically significant, positive, and direct effect on self-regulation and self-efficacy, a negative and direct effect on the intention to drop-out and a positive and direct effect on present well-being. The effect of future attitude on future well-being is positive, but indirect, as it is mediated by the other variables included in the model (especially by present well-being) [92,93].

Furthermore, concerning the second hypothesis, the results highlight the significant positive effects, direct and indirect, exerted by self-efficacy on present and future well-being [42,46].

The fourth hypothesis was confirmed by our data, highlighting the significant and negative effects of drop-out intention on present and future well-being [69].

The fifth hypothesis appears to be confirmed, consistent with what has been established in the literature [9].

Only the effects of the self-regulation variable on the remaining variables (intention to drop-out, present well-being, and future well-being) were not supported by our data (the third hypothesis). These data are not in line with the literature [49,50].

In general, our data show an important analysis of which elements should be considered to foster students' well-being. The aim of maintaining positive well-being is

fundamental to ensuring that students can have a future in terms of their overall maturation.

Although the subject of well-being is often discussed in the literature [6–10], few models have attempted to study the relationships among the variables that lead to it among university students. To the best of our knowledge, few studies have investigated the measures that universities could take to promote protective factors in the university environment. It would be interesting to focus on whether universities promote well-being programmes to help students increase their awareness of their path, above all in terms of time perspective, self-regulation, and self-efficacy. Institutions should organise more activities to address students' well-being and to develop models that ensure student well-being on multiple levels.

Despite the innovative perspective adopted in this study, with the assessment of the direct and indirect effects of the variables examined, it is necessary to recognise the limitations of the research. These limitations are related to the non-probabilistic sampling applied (related to the availability of participants) and the limited number of students involved. It would have been useful to involve a larger number of students and to expand the number of dimensions investigated to prepare a model that could consider more interactions with direct and indirect effects. Another limitation due to the chosen study design does not allow causal inferences to be made. The sample is almost exclusively composed of females and is derived from a single field of study (humanistic): hence, this does not support the generalisation of results. There are other demographic variables that may have an impact on students' well-being (e.g., living conditions, international students during the COVID-19 pandemic, students with learning disabilities, etc.) and that are not considered in the present research.

These findings might be related to specific empirical suggestions that are useful for enhancing students' well-being and professional success. It would be interesting to promote technology solutions, such as smartphone apps, to make advisors more accessible to support students in reaching out and seeking assistance [94]. Well-being support should also be empowered with platforms to conduct online well-being checks [95,96]. Research on student well-being can help administrators/educators at colleges and universities to understand the degree to which their students are self-accepting, are pursuing meaningful goals with a sense of purpose in life, have established quality ties with others, are autonomous in thought and action, can manage complex environments to suit personal needs and values and continue to grow and develop.

**Author Contributions:** Conceptualization, M.L.M., M.A., C.C., M.P.P.; methodology, M.L.M., M.A., C.C. and M.P.P.; formal analysis, M.A.; investigation, M.L.M., M.A., C.C., D.B., R.R., M.P.P.; data curation, M.L.M. and M.A.; writing—original draft preparation, M.L.M., MA, C.C., D.B., R.R., M.P.P.; writing—review and editing, M.L.M., M.A., C.C., D.B., R.R., M.P.P.; supervision, M.P.P. All authors have read and agreed to the published version of the manuscript.

**Funding:** This research received no external funding.

**Institutional Review Board Statement:** The study was conducted in accordance with the Declaration of Helsinki, and approved by the Ethics Committee of University of Cagliari (protocol code 33643_20220214, date of approval 13 February 2022).

**Informed Consent Statement:** Informed consent was obtained from all subjects involved in the study.

**Data Availability Statement:** The datasets for this study are available from the corresponding author upon reasonable request.

**Acknowledgments:** The authors would like to thank the students participating in this study.

**Conflicts of Interest:** The authors declare no conflict of interest.

# Appendix A

**Table A1.** PLS-SEM: Inner model.

| Ypothesis | Relationship | Standardized beta | Standard deviation | T-Value | *p* | Decision |
|---|---|---|---|---|---|---|
| | Path coefficient | | | | | |
| **H1A** | **STPI → Self regulation** | **0.524** | **0.065** | **7.980** | **<0.001** | **Supported** |
| **H1B** | **STPI → Self efficacy** | **0.481** | **0.063** | **7.461** | **<0.001** | **Supported** |
| **H1C** | **STPI → Intention drop out** | **−0.302** | **0.082** | **3.592** | **<0.001** | **Supported** |
| **H1D** | **STPI → Well being present** | **0.252** | **0.089** | **2.842** | **0.004** | **Supported** |
| H1E | STPI → Well being future | 0.043 | 0.049 | 0.891 | 0.373 | Not supported |
| H2A | Self efficacy → Intention drop out | −0.107 | 0.079 | 1.313 | 0.189 | Not supported |
| H2B | Self efficacy → Self regulation | −0.018 | 0.098 | 0.163 | 0.871 | Not supported |
| **H2C** | **Self efficacy → Well being present** | **0.193** | **0.090** | **2.163** | **0.031** | **Supported** |
| H2D | Self efficacy → Well being future | 0.079 | 0.048 | 1.670 | 0.095 | Not supported |
| H3A | Self regulation → Intention drop out | 0.103 | 0.093 | 1.172 | 0.241 | Not supported |
| H3B | Self regulation → Well being present | −0.056 | 0.077 | 0.798 | 0.425 | Not supported |
| H3C | Self regulation → Well being future | −0.057 | 0.047 | 1.285 | 0.199 | Not supported |
| **H4A** | **Intention drop out → Well being present** | **−0.211** | **0.068** | **3.061** | **0.002** | **Supported** |
| H4B | Intention drop out → Well being future | −0.014 | 0.048 | 0.272 | 0.786 | Not supported |
| **H5A** | **Well being present → Well being future** | **0.817** | **0.040** | **2.475** | **<0.001** | **Supported** |
| | Total indirect effects | | | | | |
| **H1** | **STPI → Well being future** | **0.323** | **0.065** | **4.895** | **<0.001** | **Supported** |
| H1 | STPI → Well being present | 0.128 | 0.075 | 1.605 | 0.109 | Not supported |
| H1 | STPI → Intention drop out | 0.003 | 0.058 | 0.112 | 0.911 | Not supported |
| H1 | STPI → Self regulation | −0.007 | 0.048 | 0.159 | 0.873 | Not supported |
| **H2** | **Self efficacy → Well being future** | **0.179** | **0.075** | **2.397** | **0.017** | **Supported** |
| H2 | Self efficacy → Well being present | 0.024 | 0.021 | 1.086 | 0.278 | Not supported |
| H2 | Self efficacy → Intention drop out | 0.000 | 0.013 | 0.136 | 0.892 | Not supported |
| H3 | Self regulation → Well being future | −0.065 | 0.065 | 1.069 | 0.285 | Not supported |
| H3 | Self regulation → Well being present | −0.021 | 0.021 | 1.076 | 0.282 | Not supported |
| **H4** | **Intention drop out → Well being future** | **−0.172** | **0.055** | **3.098** | **0.002** | **Supported** |
| | Specific indirect effects | | | | | |
| H1 | STPI → Intention drop out → Well being future | 0.004 | 0.015 | 0.253 | 0.800 | Not supported |

**Table A1.** *Cont.*

| Ypothesis | Relationship | Standardized beta | Standard deviation | T-Value | *p* | Decision |
|---|---|---|---|---|---|---|
| H1 | STPI → Self regulation → Well being present → Well being future | −0.024 | 0.034 | 0.761 | 0.446 | Not supported |
| H1 | STPI → Self efficacy → Self regulation → Well being future | 0.000 | 0.003 | 0.133 | 0.894 | Not supported |
| H1 | STPI → Self efficacy → Intention drop out → Well being present | 0.011 | 0.010 | 1.045 | 0.296 | Not supported |
| H1 | STPI → Self efficacy → Well being future | 0.038 | 0.025 | 1.547 | 0.122 | Not supported |
| **H1** | **STPI → Self efficacy → Well being present** | **0.093** | **0.046** | **1.994** | **0.046** | **Supported** |
| H1 | STPI → Self efficacy → Self regulation | −0.007 | 0.048 | 0.159 | 0.873 | Not supported |
| H1 | STPI → Self efficacy → Self regulation → Intention drop out → Well being future | 0.000 | 0.000 | 0.032 | 0.974 | Not supported |
| H1 | STPI → Self regulation → Intention drop out → Well being future | −0.001 | 0.004 | 0.197 | 0.844 | Not supported |
| H1 | STPI → Self efficacy → Self regulation → Intention drop out → Well being present → Well being future | 0.000 | 0.001 | 0.124 | 0.901 | Not supported |
| H1 | STPI → Self efficacy → Self regulation → Well being present → Well being future | 0.000 | 0.003 | 0.108 | 0.914 | Not supported |
| H1 | STPI → Self efficacy → Intention drop out → Well being future | 0.001 | 0.003 | 0.201 | 0.841 | Not supported |
| H1 | STPI → Self efficacy → Intention drop out → Well being present → Well being future | 0.009 | 0.008 | 1.053 | 0.292 | Not supported |
| H1 | STPI → Self efficacy → Self regulation → Well being present | 0.000 | 0.004 | 0.109 | 0.913 | Not supported |
| H1 | STPI → Self efficacy → Self regulation → Intention drop out | 0.000 | 0.006 | 0.132 | 0.895 | Not supported |
| **H1** | **STPI → Intention drop out → Well being present → Well being future** | **0.052** | **0.023** | **2.192** | **0.028** | **Supported** |
| **H1** | **STPI → Well being present → Well being future** | **0.206** | **0.074** | **2.810** | **0.005** | **Supported** |
| H1 | STPI → Self regulation → Intention drop out → Well being present → Well being future | −0.009 | 0.009 | 1.035 | 0.301 | Not supported |
| H1 | STPI → Self efficacy → Self regulation → Intention drop out → Well being present | 0.000 | 0.001 | 0.123 | 0.902 | Not supported |
| H1 | STPI → Self regulation → Intention drop out → Well being present | −0.011 | 0.011 | 1.024 | 0.306 | Not supported |

**Table A1.** *Cont.*

| Ypothesis | Relationship | Standardized beta | Standard deviation | T-Value | *p* | Decision |
|---|---|---|---|---|---|---|
| H1 | STPI → Self regulation → Intention drop out | 0.055 | 0.051 | 1.108 | 0.268 | Not supported |
| **H1** | **STPI → Intention drop out → Well being present** | **0.064** | **0.028** | **2.164** | **0.031** | **Supported** |
| H1 | STPI → Self regulation → Well being future | −0.030 | 0.025 | 1.227 | 0.220 | Not supported |
| H1 | STPI → Self regulation → Well being present | −0.030 | 0.041 | 0.769 | 0.442 | Not supported |
| **H1** | **STPI → Self efficacy → Well being present → Well being future** | **0.076** | **0.038** | **1.999** | **0.046** | **Supported** |
| H1 | STPI → Self efficacy → Intention drop out | −0.051 | 0.039 | 1.255 | 0.210 | Not supported |
| H2 | Self efficacy → Self regulation → Intention drop out → Well being present | 0.000 | 0.003 | 0.127 | 0.899 | Not supported |
| H2 | Self efficacy → Intention drop out → Well being present → Well being future | 0.019 | 0.016 | 1.110 | 0.267 | Not supported |
| H2 | Self efficacy → Self regulation → Intention drop out | 0.000 | 0.013 | 0.136 | 0.892 | Not supported |
| H2 | Self efficacy → Self regulation → Intention drop out → Well being future | 0.000 | 0.001 | 0.033 | 0.973 | Not supported |
| H2 | Self efficacy → Intention drop out → Well being present | 0.023 | 0.019 | 1.104 | 0.270 | Not supported |
| H2 | Self efficacy → Intention drop out → Well being future | 0.001 | 0.007 | 0.207 | 0.836 | Not supported |
| H2 | Self efficacy → Self regulation → Well being present | 0.001 | 0.009 | 0.109 | 0.913 | Not supported |
| H2 | Self efficacy → Self regulation → Well being present → Well being future | 0.001 | 0.007 | 0.109 | 0.913 | Not supported |
| **H2** | **Self efficacy → Well being present → Well being future** | **0.157** | **0.073** | **2.160** | **0.031** | **Supported** |
| H2 | Self efficacy → Self regulation → Well being future | 0.001 | 0.007 | 0.135 | 0.893 | Not supported |
| H2 | Self efficacy → Self regulation → Intention drop out → Well being present → Well being future | 0.000 | 0.002 | 0.129 | 0.897 | Not supported |
| | | | | | | Not supported |
| H3 | Self regulation → Intention drop out → Well being present → Well being future | −0.017 | 0.017 | 1.086 | 0.277 | Not supported |
| H3 | Self regulation → Intention drop out → Well being present | −0.021 | 0.021 | 1.076 | 0.282 | Not supported |
| H3 | Self regulation → Well being present → Well being future | −0.046 | 0.063 | 0.790 | 0.430 | Not supported |

**Table A1.** *Cont.*

| Ypothesis | Relationship | Standardized Beta | Standard Deviation | T-Value | *p* | Decision |
|---|---|---|---|---|---|---|
| H3 | Self regulation → Intention drop out → Well being future | −0.001 | 0.007 | 0.207 | 0.836 | Not supported |
| **H4** | **Intention drop out → Well being present → → Well being future** | **−0.172** | **0.055** | **3.098** | **0.002** | **Supported** |
| | Total effects | | | | | |
| **H1** | **STPI → Self efficacy** | **0.481** | **0.063** | **7.461** | **<0.001** | **Supported** |
| **H1** | **STPI → Well being future** | **0.366** | **0.072** | **5.036** | **<0.001** | **Supported** |
| **H1** | **STPI → Well being present** | **0.380** | **0.066** | **5.700** | **<0.001** | **Supported** |
| **H1** | **STPI → Intention drop out** | **−0.298** | **0.064** | **4.487** | **<0.001** | **Supported** |
| **H1** | **STPI → Self regulation** | **0.517** | **0.054** | **9.358** | **<0.001** | **Supported** |
| **H2** | **Self efficacy → Well being future** | **0.258** | **0.095** | **2.735** | **0.006** | **Supported** |
| **H2** | **Self efficacy → Well being present** | **0.217** | **0.090** | **2.397** | **0.017** | **Supported** |
| H2 | Self efficacy → Intention drop out | −0.107 | 0.078 | 1.343 | 0.179 | Not supported |
| H2 | Self efficacy → Self regulation | −0.018 | 0.098 | .163 | 0.871 | Not supported |
| H3 | Self regulation → Well being future | −0.121 | 0.080 | 1.621 | 0.105 | Not supported |
| H3 | Self regulation → Well being present | −0.077 | 0.079 | 1.061 | 0.289 | Not supported |
| H3 | Self regulation → Intention drop out | 0.103 | 0.093 | 1.172 | 0.241 | Not supported |
| **H4** | **Intention drop out → Well being future** | **−0.186** | **0.068** | **2.668** | **0.008** | **Supported** |
| **H4** | **Intention drop out → Well being present** | **−0.211** | **0.068** | **3.061** | **0.002** | **Supported** |
| **H5** | **Well being present → Well being future** | **0.817** | **0.040** | **2.475** | **<0.001** | **Supported** |

Note: H1a, hypothesis 1a; H1b, hypothesis 1b; H2a, hypothesis 2a; H2b, hypothesis 2b; H3a, hypothesis 3a, H3b: hypothesis 3b; p, probability. The bold type highlights the significant effects.

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
