# Peer review of "Present and Future Undergraduate Students’ Well-Being: Role of Time Perspective, Self-Efficacy, Self-Regulation and Intention to Drop-Out"

_education, doi:10.3390/educsci13020202_

Round 1

Reviewer 1 Report

Dear Authors,

After reviewing the article, I will proceed to list some comments so that you can improve its quality, since I consider it potentially suitable for publication if the recommendations are heeded:

First, on the theoretical framework:

-          I consider it necessary to deepen more on the relationship between self-regulation and well-being. As you know, self-regulation of learning is a complex and multifactorial construct composed of multiple areas. I believe it is necessary to delve deeper at the theoretical level into the relationship between these areas and well-being. Especially, considering that this aspect is referred to in the discussion.

-          In general, the relationship between the variables and welfare is explained in a very superficial way; it is necessary to explain and deepen this relationship. In the same way, it would be necessary to relate them to each other and not only to well-being to provide more robustness to the research. Framing it in a more solid and integrated theoretical framework. Especially considering that in the hypotheses and in the graphical representation provided, this relationship is exposed.

Second, on the materials and methods:

-          It is advisable to describe the procedure and the methodology used explicitly.

-          The mean age is in parentheses along with the standard deviation, both symbols in italics.

-          I do not consider it relevant to include all the mean and standard deviation of all the answers. I do consider it necessary to include the frequency of gender, age, and education, but in a redacted form, the table is too long.

-          I advise presenting the instruments in the same order in which the variables are presented in the hypotheses or in the theoretical framework.

In third place, in relation to results:

-          Regarding table 2. I would try to write this information in some summarized form or highlighting only the most salient indicators. If it is possible to include it as extra material or annex, I would do so but not in the main part of the document. The information is not necessary to include such a large table, it should be mentioned but not in this way.

-          The same thing happens with table 5. If you have already explained the most relevant results, the table is duplicating information and including a lot of other information that is not relevant. For this reason, you must make a choice: either you continue explaining it in a slightly more written way, or you can leave the table out... If there is an editorial way to put it as extra content in case the reader demands it, great, otherwise I advise you to try not to repeat the information in two different ways. As you will understand, these are very large tables that are not functional for the article.

Fourth, regarding the discussion and conclusions

-          For example, section 4 is at the end of the document and starts on the next page. It does not look very aesthetic.

-          I note a lack of discussion of the results with previous research. The results should be referred to the previous research to affirm that it is in line or that it is an alternative result. In conjunction with this, the result should be justified by providing a concrete explanation for the result.

-          I would recommend reviewing the theoretical frameworks of Edward F. Diener and Carol Ryff as reference authors in the field of both subjective and psychological well-being to support the construct. I think it may be helpful especially when relating it to the other variables in this study.

Finally, I notice inconsistencies in the format of the references. Some are in APA format and others in Chicago or Harvard format. In the same way I find mistakes in the different formats, lack of information such as the year of publication, among others. For example, we can see:

·         Seligman, M. PERMA and the Building Blocks of Well-Being. J Posit Psychol 2018, 13, 333–335, 457 doi:10.1080/17439760.2018.1437466

·         Tinto, V. (1975). Dropout from Higher Education: A Theoretical Synthesis of Recent Research. Review of 487 Educational Research, 45(1), 89–125. https://doi.org/10.3102/00346543045001089

·         Putting Time in Perspective: A Valid, Reliable Individual-Differences Metric. In: Stolarski, M., Fieulaine, N., 506 van Beek, W. (eds) Time Perspective Theory; Review, Research and Application. Springer, Cham. 507 https://doi.org/10.1007/978-3-319-07368-2_2

I hope you will be able to attend to these considerations, which are always aimed at improvement.

With my best wishes.

Author Response

Dear Reviewers,

Thank you for your letter Ref. Manuscript ID: Education-2132098-R1, entitled “Present and Future Undergraduate Students’ Well-being: the Role of Time Perspective, Self-efficacy, Self-regulation and Intention to Drop-out”, and for giving us the opportunity to review and resubmit the paper.

We are very grateful to your and the reviewers’ comments and suggestions; we are deeply appreciative of your careful reading.

Detailed replies to your comments are enumerated below, with the list of modifications and integrations. We hope this revised version now satisfies the requirements for publication in your journal.

Then, we submit the revised version of paper; for clarity new portions, added or modified in response to the referees’ comments, are highlighted in the manuscript; furthermore, the tracked version of the manuscript is attached.

Thank you very much

The Authors

Reviewer 2 Report

The present study aims at exploring the students’ perception of well-being and its correlation with some variables- namely time perspective, self-efficacy, self-regulation and intention to drop-out -by attempting to build a model able to explain the interplay of these factors. The Authors recruited 192 students attending university courses leading to bachelor’s and master’s degree at the University of Cagliari (Italy). Participants were administered multiple validated scales to identify predictors of students’ present and future well-being and the data obtained were analysed using a relevant approach such as the partial least squares structural equation model, strength of this article. 

The topic is interesting and extremely timely, as the construct of well-being is crucial in our society, especially in the time of COVID-19 pandemic. University students represent a population in a vulnerable stage of life, which is why it is important to understand how facets of well-being correlate and can influence the course of study. The present research yielded to significant findings and, thus, opens to the possibility of early identification of factors influencing students’ well-being, representing a meaningful added value to the current literature. 

From a formal perspective, the language is sometimes unclear or has grammatical flaws: revision by a native speaker or from a text-revision service can improve understandability. From a methodological point of view, the introduction and discussionsections appear to be properly structured, while the methods and results sections could be improved and supplemented (see for this purpose the STROBE statement https://www.strobe-statement.org).

More in details, there are some major and minor issues that affect the general quality of the article and that the Authors may be willing to address.

Major:

·       Abstract. According to the STROBE statement, the study design should be explicitly mentioned in this section. 

·       Materials and Methods, Participants. Demographic characteristics of the sample represent a result of the study. Therefore, this paragraph should be moved to the appropriate section, along with Table 1, as reported it the STROBE statement. 

·       Materials and Methods. The first part of this section should provide a detailed description of study procedures, in particular of the study design and recruitment path (see the STROBE statement). To enable a proper understanding of methodological aspects, I recommend that the Authors add this essential information. 

·       Materials and Methods, Data analyses. The statistical analyses performed should be further explained in this paragraph. To align with the recommendations of the STROBE statement, I suggest that the Authors expand further on what is declared in this paragraph by better describing the "two-stage analysis approach" and by clearly indicating the level of statistical significance considered.

·       Discussion, LL 421-427.  In general, this section appears well structured, providing a fair summary and commentary of the results. Nevertheless, the contextualisation of the results in the existing literature is somewhat sparse and could be improved. With this in mind, there are several articles that may be considered relevant in dealing with such a vast and topical issue as the wellbeing of university students, both in terms of preventive factors and applicable interventions. In particular, I would suggest implementing some references in these lines with respect to the topics mentioned. See, for example, the following articles:  

-     doi: 10.1371/journal.pone.0266725

-     doi: 10.1016/j.rpsm.2022.04.005

-     doi: 10.3389/fpubh.2022.924711

-     doi: 10.3390/brainsci12091155

·       Limitations. I would suggest implementing this paragraph by adding another 3 limitations: 

-    I would recommend making it explicit that the chosen study design does not allow causal inferences to be made.

-    The sample is almost exclusively composed of females and is derived from a single field of study (humanistic): hence, this does not support the generalisation of results, and this limitation should be clearly stated.

-    There are other demographic variables that may have an impact on students’ well-being (e. g. living conditions, international students during COVID-19 pandemic) and that are not considered in the present research, therefore, limiting its explorative value. 

Minor:

·       Tables. As a general comment, the tables appear uneven in form, layout, and content. I would advise the Authors to revise them and make them more usable for the reader, thus improving the understanding of the results of their research.

·       Results, Figure 2. I reckon that the use of visual components can be useful in highlighting which interactions are significant and which are not. Using arrows of different colour or thickness can facilitate the visual understanding of the study results.

·       Results, Table 5. The footnote states “The green ink highlights the significant effects”. Since the tables are shown without colours, I would recommend adapting the table to the Journal indications.

·       Results, Structural Model, LL 366. The word “effects” is missing.

·       Discussion, LL 436-438. Several syntheses of evidence on this topic can be found in literature, which offer interesting insights. It might be interesting to mention some of them, including, for example, the one by Ferrari and colleagues (doi: 10.2196/39686).

·       As a general comment, references in the text should be cited at the end of the sentence, whereas references’ format in the reference list should be coherent in the whole section. I recommend the Authors to check them for accuracy and formal consistency (see for this purpose MDPI Instruction for Authors https://www.mdpi.com/journal/education/instructions#preparation).

Author Response

Dear Editor and Referees,

Thank you for your letter Ref. Manuscript ID: Education-2132098-R1, entitled “Present and Future Undergraduate Students’ Well-being: the Role of Time Perspective, Self-efficacy, Self-regulation and Intention to Drop-out”, and for giving us the opportunity to review and resubmit the paper.

We are very grateful to your and the reviewers’ comments and suggestions; we are deeply appreciative of your careful reading.

Detailed replies to your comments are enumerated below, with the list of modifications and integrations. We hope this revised version now satisfies the requirements for publication in your journal.

Then, we submit the revised version of paper; for clarity new portions, added or modified in response to the referees’ comments, are highlighted in the manuscript; furthermore, the tracked version of the manuscript is attached.

Thank you very much

The Authors

Round 2

Reviewer 1 Report

Dear Authors,
After reviewing the changes made and the responses you have provided, I understand that the changes have been addressed with sufficient quality to accept the article for publication.
Congratulations to the authors
With best wishes,

Author Response

Thank you for all your precious suggestions and for your appreciation.

Best regards

The authors

Reviewer 2 Report

I would first like to thank the Authors for their valuable work in editing the manuscript according to my comments and clarifying some methodological concerns. The quality of this manuscript has significantly improved, both in terms of comprehensibility and scientific soundness. As a general comment, the language is much improved, although I would suggest that the Authors check for misspells or inaccuracies.

There are a few minor issues I would like to bring to the attention of the Authors.

·       Abstract, lines 10-11. The sentence “The aim of this study is how the interaction…” should be completed with an infinitive form verb. 

·       Abstract, lines 16-17. “A partial least squares structural equation modeling (PLS-SEM) analysis was carried out to examine among the construct variables examined herein and well-being”. The latter part of this sentence is not clear, I would recommend that the Authors reformulate this line. I would also suggest to accurately check the spelling of the word “modelling”. 

·       Abstract. The type of study design chosen, as stated in the STROBE statement (https://www.strobe-statement.org), should be made explicit in this section.

·       Results, lines 373-374. The Authors should replace the word “receptively” with the word “respectively”.

Author Response

Thanks for your words. Thank you for all your precious suggestions and for your appreciation.

Best regards

The authors
